# Semi-intact ex vivo approach to investigate spinal somatosensory circuits

**Junichi Hachisuka[1,2], Kyle M Baumbauer[1,2†], Yu Omori[1,2], Lindsey M Snyder[1,2], H Richard Koerber[1,2]\*, Sarah E Ross[1,2]\***

[1]Department of Neurobiology, University of Pittsburgh, Pittsburgh, United States; [2]Pittsburgh Center for Pain Research, University of Pittsburgh, Pittsburgh, United States

**Abstract** The somatosensory input that gives rise to the perceptions of pain, itch, cold and heat are initially integrated in the superficial dorsal horn of the spinal cord. Here, we describe a new approach to investigate these neural circuits in mouse. This semi-intact somatosensory preparation enables recording from spinal output neurons, while precisely controlling somatosensory input, and simultaneously manipulating specific populations of spinal interneurons. Our findings suggest that spinal interneurons show distinct temporal and spatial tuning properties. We also show that modality selectivity — mechanical, heat and cold — can be assessed in both retrogradely labeled spinoparabrachial projection neurons and genetically labeled spinal interneurons. Finally, we demonstrate that interneuron connectivity can be determined via optogenetic activation of specific interneuron subtypes. This new approach may facilitate key conceptual advances in our understanding of the spinal somatosensory circuits in health and disease.

**\*For correspondence:** rkoerber@ pitt.edu (HRK); saross@pitt.edu (SER)

**Present address:** [†]School of Nursing, The Center for Advancing Management of Pain, University of Connecticut, Storrs, United States

**Competing interests:** The authors declare that no competing interests exist.

## Introduction

Noxious chemical, mechanical and thermal stimuli are conveyed to the dorsal horn of the spinal cord where they are integrated by excitatory and inhibitory interneurons. Under some circumstances, these neural circuits reduce the degree to which noxious input is conveyed to the brain (*Gebhart, 2004*; *Moayedi and Davis, 2013*). In others, the somatosensory input is amplified by spinal circuits, leading to allodynia and hyperalgesia, two hallmarks of chronic pain (*Sandkühler, 2009*). Given the central role of the spinal dorsal horn in the modulation of noxious input, there is a great need for an improved understanding of the underlying spinal microcircuits and the computations they perform.

Electrophysiological recording from spinal neurons is one of the best approaches to study the functional connectivity among spinal neurons in detail. Historically, such studies were performed using in vivo recordings (*Brown et al., 1987*; *Carstens, 1997*; *Craig et al., 2001*; *Dickenson and Sullivan, 1986*; *Jinks and Carstens, 2000, 1999*; *Koerber et al., 1991*; *Randic et al., 1988*; *Torsney and Fitzgerald, 2002*; *Urch and Dickenson, 2003*; *Zhang et al., 2006*). However, a considerable limitation to this approach is that it is difficult to target known cell types in the dorsal horn. With the advent of a new generation of molecular genetic tools, it has been possible to overcome this constraint by recording from genetically defined subsets of neurons (*Bourane et al., 2015a, 2015b*; *Cui et al., 2016*; *Duan et al., 2014*; *Hantman et al., 2004*; *Heinke et al., 2004*; *Kardon et al., 2014*; *Peirs et al., 2015*; *Petitjean et al., 2015*; *Zeilhofer et al., 2005*). Generally, though, these experiments have been performed within the context of a spinal slice, which suffers from the shortcoming that key components of the nociceptive circuit, such as the input, are severed. In this context, spinal circuits are evaluated using electrical stimulation of the whole dorsal roots, even though such a stimulus is not physiological. These issues underscored the need for an

electrophysiological approach that allows visual access for targeted recordings, and includes innervated skin for natural stimulation.

Here, we describe a new approach that achieves these goals, enabling precise determination of spinal output, as well as modality-specific control over somatosensory input, and the ability to manipulate genetically defined interneuron populations within a nociceptive circuit. First, we reveal how excitatory and inhibitory inputs can be distinguished from one another, and use this approach to characterize lamina I neurons based on temporal and spatial tuning properties. Next, we analyze modality selectivity of lamina I spinoparabrachial (SPB) projection neurons and genetically labeled spinal interneurons. Finally, we investigated the feasibility of optogenetic manipulation in the dorsal horn, and use this approach to begin mapping spinal circuitry. Together, our findings provide evidence that the semi-intact preparation is a powerful new approach that will facilitate the study of spinal somatosensory processing.

## Results

### Ex vivo somatosensory preparation

Our goal was to develop a mouse preparation that preserves intact spinal circuitry, allows physiological stimulation, and enables whole cell recordings from known neuronal subtypes in the dorsal horn. Previously, we developed an ex vivo preparation to record from primary afferents while providing natural stimulation to the skin (*Baumbauer et al., 2015*; *Jankowski et al., 2009*; *Lawson et al., 2008*; *McIlwrath et al., 2007*). Here, we extended the use of this preparation by recording from neurons in the superficial dorsal horn rather than the dorsal root ganglia. This preparation comprises of a large portion of spinal cord (~C2 – S3), together with lumbar 2 (L2) and L3 roots, ganglia, saphenous nerve and hindlimb skin (including hindpaw), dissected in continuum (*Figure 1A and B*). Spinal segments L2 and L3, which are targeted by the saphenous nerve, are regions of the spinal cord that are well-suited for visually-guided whole cell recordings because they contain regions of grey matter that are clearly visible between gaps in the roots (*Figure 1C*, dotted ovals). In particular, the lateral aspect of the dorsal horn is favorable because the white matter in this region is very thin (*Figure 1D*, green shaded area). For optimal stimulation of the skin, we created a custom recording chamber in which the skin rests on a perforated platform at the air-liquid interface, allowing the dermis to be bathed with oxygenated aCSF, while the stratum corneum is exposed and thus accessible to mechanical and thermal stimuli (*Figure 1E*). In addition, to accommodate pharmacological studies, we also designed an alternative, small volume chamber with directed flow optimized for drug application (*Figure 1—figure supplement 1A and B*). The final challenge was to visualize spinal neurons for recordings. To this end, we employed side illumination with a high-powered infrared light emitting diode, which allowed visualization and patch-clamp recording of cells up to 50 µm from the surface of the spinal cord (*Figure 1F*), consistent with previous reports (*Safronov et al., 2007*; *Szucs et al., 2009*).

### Temporal and spatial response analysis of excitatory and inhibitory input

One of the key advantages of this new approach is that it facilitates whole cell recordings in the context of a relatively intact preparation. This is beneficial because understanding neuronal function depends on measuring synaptic input (sub-threshold excitatory and inhibitory synaptic currents) and neuronal output (action potentials). Previous studies, however, have generally not been performed in a manner that enables the detailed analysis of sub-threshold inputs upon stimulation of the skin. In contrast, our preparation allows us to combine voltage clamp mode to characterize synaptic inputs onto the recorded cell, as well as current clamp mode to measure the firing evoked by natural cutaneous stimuli.

To facilitate the study of both excitatory post-synaptic currents (EPSCs) and inhibitory post-synaptic currents (IPSCs) in the same cells, we took advantage of the fact that at the reversal potential for chloride ($V_H = -70$ mV) only EPSCs are observed, whereas at a depolarized holding potential ($V_H = -40$ mV) it is possible to clearly resolve IPSCs as outward current (*Yoshimura and Nishi, 1993*). Previously, Yoshimura and colleagues analyzed the synaptic currents in lamina II neurons that are

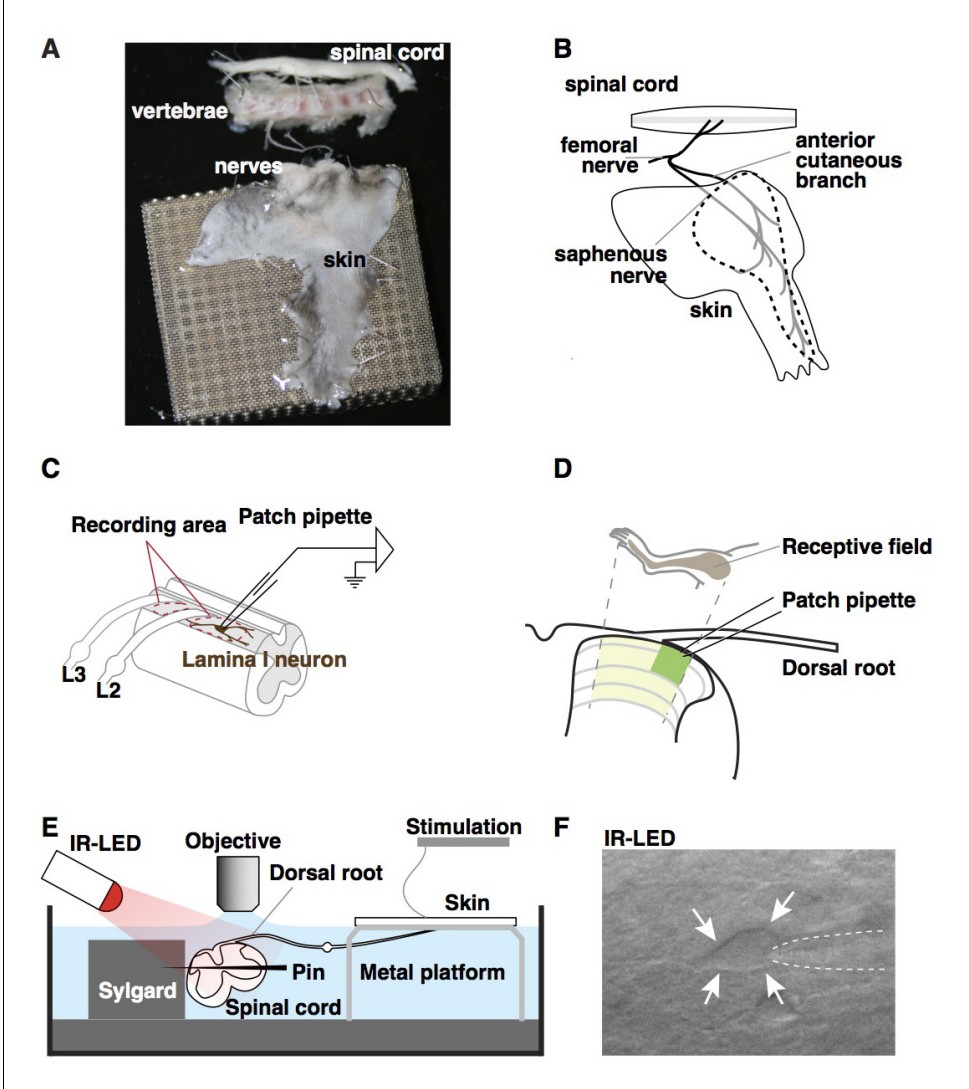

**Figure 1.** Ex vivo semi-intact somatosensory preparation. (A–B) Photograph (A) and schematic (B) of the semi-intact somatosensory preparation. The right dorsal hindlimb skin with saphenous nerve and anterior cutaneous branch are dissected in continuity with L2 and L3 roots and the spinal cord. Region containing receptive fields of recorded neurons is dotted. (C) Diagram illustrating targeted region of spinal cord. Patch-clamp recordings are made rostral to L2 root or between L2 and L3 (red dotted circles). (D) Somatotopic organization of saphenous nerve and anterior cutaneous branch. Hindlimb skin is represented in the area of spinal cord that is shaded yellow. The region that is most accessible for recording is shaded green. (E) Side view of the preparation. The spinal cord is pinned to Sylgard chamber and illuminated by high-power infrared LED. (F) Oblique infrared LED illumination reveals a lamina I neuron, as indicated by arrows, which is targeted for recording by glass electrode, indicated with dotted lines.

The following figure supplement is available for figure 1:

**Figure supplement 1.** Alternative recording chamber optimized for pharmacological experiments.

evoked by cutaneous stimulation (*Furue et al., 1999*; *Narikawa et al., 2000*); however, the input onto lamina I neurons remained uncharacterized.

In an effort to test the functionality of our new approach, we began by analyzing synaptic currents onto lamina I neurons in response to mechanical stimulation, using a small paintbrush, which elicits activity in both nociceptors and non-nociceptors. We found that approximately half of lamina I

neurons showed synaptic currents upon brushing of the anterior hindlimb skin. Of these, brush-evoked EPSCS and IPSCS (*Figure 2A and D*) were observed in one quarter of responding lamina I neurons. In contrast, only EPSCs were observed upon brush stimulation for approximately half of lamina I neurons (*Figure 2B and D*), whereas only IPSCs were observed in the remaining quarter (*Figure 2C and D*). These data show that the somatosensory preparation can be used to visualize excitatory and inhibitory inputs evoked by natural stimulation.

Next, we investigated spatial properties of inhibitory and excitatory receptive fields of lamina I neurons, analyzing action potentials, EPSCs and IPSCs. A von Frey filament (2 g) was used to facilitate the identification of the receptive field border. A typical example is shown, illustrating a neuron whose inhibitory receptive field was significantly larger than its excitatory receptive field (*Figure 2E*). Brushing the center of the receptive field induced EPSCs (upper trace) at a holding potential of −70 mV, as well as action potentials and IPSPs in current clamp mode (middle trace). In contrast, stimulation of the off-center area evoked only IPSPs, but no EPSPs (bottom trace). Of lamina I neurons that receive both excitatory and inhibitory inputs upon stimulation of the skin, the majority (5 of 6) show this type of center-surround receptive field (*Figure 2F*), consistent with previous intracellular recordings in cat (*Brown et al., 1987*; *Koerber et al., 2006*; *Koerber and Mirnics, 1996*).

Finally, we examined the temporal response of lamina I neurons upon mechanical stimulation of the skin with a von Frey filament (2 g). Most lamina I neurons (48 of 61) showed EPSCs only during the period of mechanical stimulation (*Figure 2G and I*). However, approximately one-fifth showed a very distinct temporal response in which a prolonged afterdischarge of EPSCs and action potentials outlasted the stimulus duration by several seconds (*Figure 2H and I*), consistent with previous findings from recordings in vivo (*Woolf and Fitzgerald, 1983*). Taken together, these experiments not only show feasibility of the semi-intact somatosensory preparation for the analysis of spatial and temporal responses, but also provide the first detailed analysis of the synaptic inputs received by lamina I neurons upon stimulation of the skin, thereby complementing and extending previous extracellular recordings in cat, rat and monkey (*Craig et al., 2001*; *Craig and Andrew, 2002*; *Davidson et al., 2008*; *Jinks and Carstens, 1999*; *Zhang et al., 2006*).

## Characterization of lamina I projection neurons

While the basic characterization of lamina I neurons to natural input is important, our ability to interpret these experiments was hampered because the identity of the recorded cell remained unknown. We therefore sought to determine the feasibility of recording from known cell types. Spinal projection neurons were of particular interest to us, since they are the output neurons that convey nociceptive information to the brain. However, while many spinal projection neurons are located in lamina I, they can be difficult to target since they are quite sparse, representing only 5% of total neurons in this lamina (*Todd, 2010*). Previous work in rat has revealed that most anterolateral tract projection neurons send their axons to the lateral parabrachial nucleus in the brainstem (*Al-Khater and Todd, 2009*). We therefore injected DiI into this nucleus to retrogradely label neurons SPB projection neurons, which were readily identified four days later and targeted for whole cell patch clamp recordings (*Figure 3A*). Next, we characterized the modality selectivity by assessing the response to mechanical force (M; 1 g von Frey filament), heat (H; 50°C saline), and cold (C; 0°C saline) stimuli. These experiments revealed the existence of several discrete functional classes of SPB projection neurons. Half of these neurons were considered to be polymodal because action potentials were evoked in response to both mechanical and thermal stimulation (*Figure 3B, C and F*). The other half appeared tuned to thermal stimuli, responding to either cold only (*Figure 3D*) or cold and heat (*Figure 3E*) as summarized (*Figure 3F*). Thus, our data show that the somatosensory preparation can be used to determine stimulus-response properties, and reveal for the first time that mouse spinal projection neurons in lamina I, like those in cat, rat and monkey, show modality-selective tuning (*Craig et al., 2001*; *Davidson et al., 2008*; *Zhang et al., 2006*).

## Recording from genetically labeled neurons

While a small number of long-range projection neurons can be labeled by virtue of their targets, the vast majority of neurons in the dorsal horn are interneurons with local projections, thus precluding this labeling approach (*Todd, 2010*). Recently, several Cre and/or Flp alleles have been used to target genetically defined populations of spinal interneurons and investigate their role through loss- or

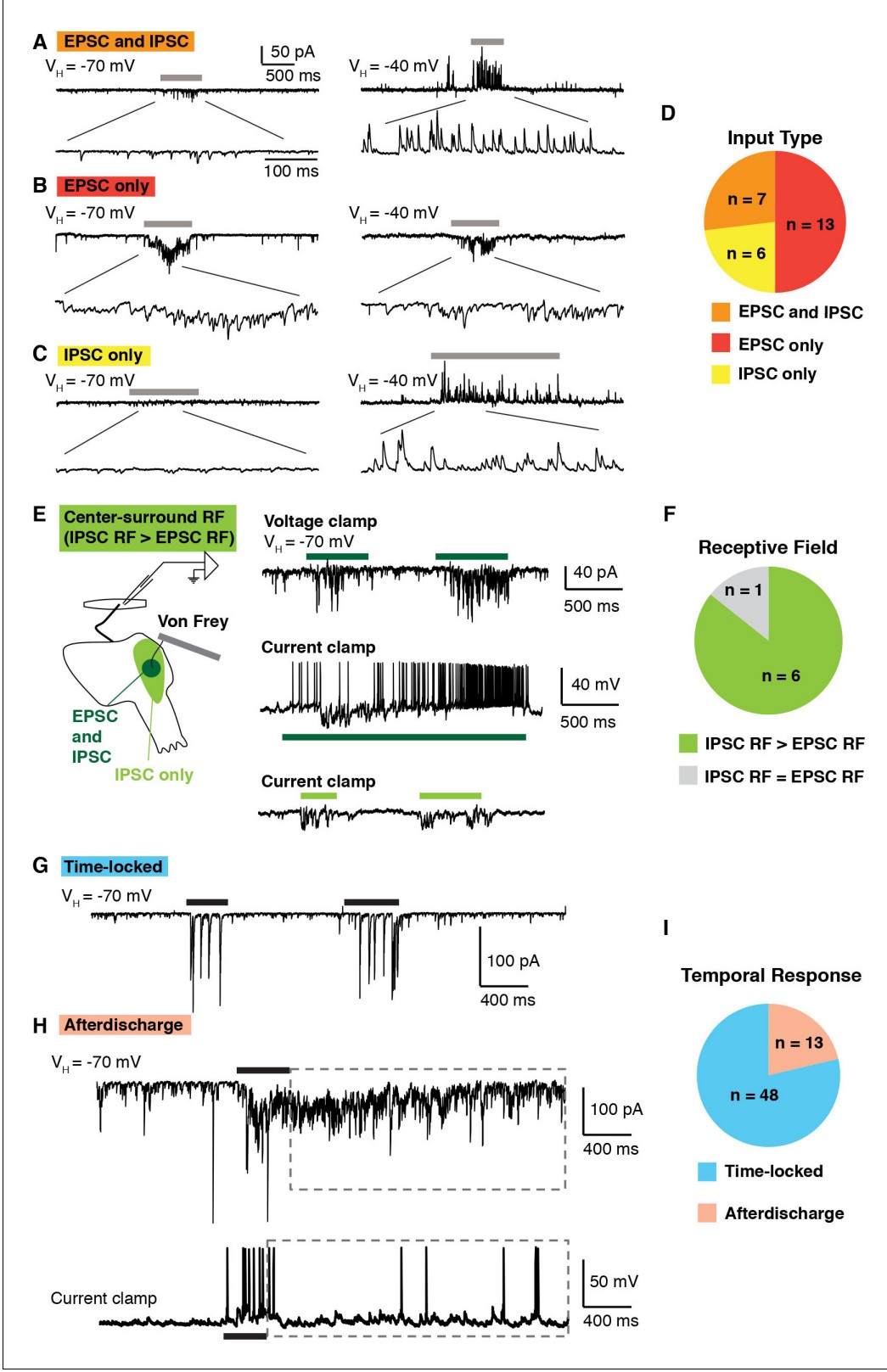

**Figure 2.** Temporal and spatial response of excitatory and inhibitory inputs. (**A–D**) Analysis of excitatory and inhibitory input onto lamina I neurons upon brushing of the skin (grey bars). At a holding potential of −70 mV, only EPSCs are observed, whereas at a holding potential of −40, IPSCs are clearly visible as outward current. Example of a lamina I cell that receives EPSCs and IPSCs (**A**), EPSCS only (**B**) and IPSCs only (**C**). Summary (**D**) of input type
*Figure 2 continued on next page*

*Figure 2 continued*

from 26 lamina I neurons. (E–F) Spatial analysis of the receptive fields of lamina I neurons. Example of a neuron that shows center-surround organization (E). Stimulation (von Frey filament 2 g) in the center of the receptive field (dark green) elicited EPSCs in voltage clamp at V$_H$ of −70 mV (top trace). In current clamp, mechanical stimulation causes IPSPs as well as action potentials (middle trace). Mechanical stimulation on the outer side of the receptive field (light green area) caused IPSPs but no EPSPs (bottom trace). Summary (F) of receptive fields from seven lamina I neurons. 6 of 7 showed inhibitory receptive fields that were significantly larger than their excitatory receptive fields. (G–I) Temporal analysis of responses of lamina I neurons. Example of a neuron that showed time-locked responses (G) and afterdischarge (boxed dashed line; H) to mechanical stimulation (von Frey filament, 2 g, black bar) of the receptive field. Summary (I) of temporal responses of 61 lamina I neurons.

gain-of-function approaches in vivo (*Bourane et al., 2015a*, *2015b*; *Cui et al., 2016*; *Duan et al., 2014*; *Peirs et al., 2015*; *Petitjean et al., 2015*). Less is known, however, about the specific functions of these cells within the context of a spinal microcircuit. We therefore investigated whether the semi-intact somatosensory preparation could be used to record from genetically defined interneurons and characterize their modality tuning. Because the genetically-labeled cells were located

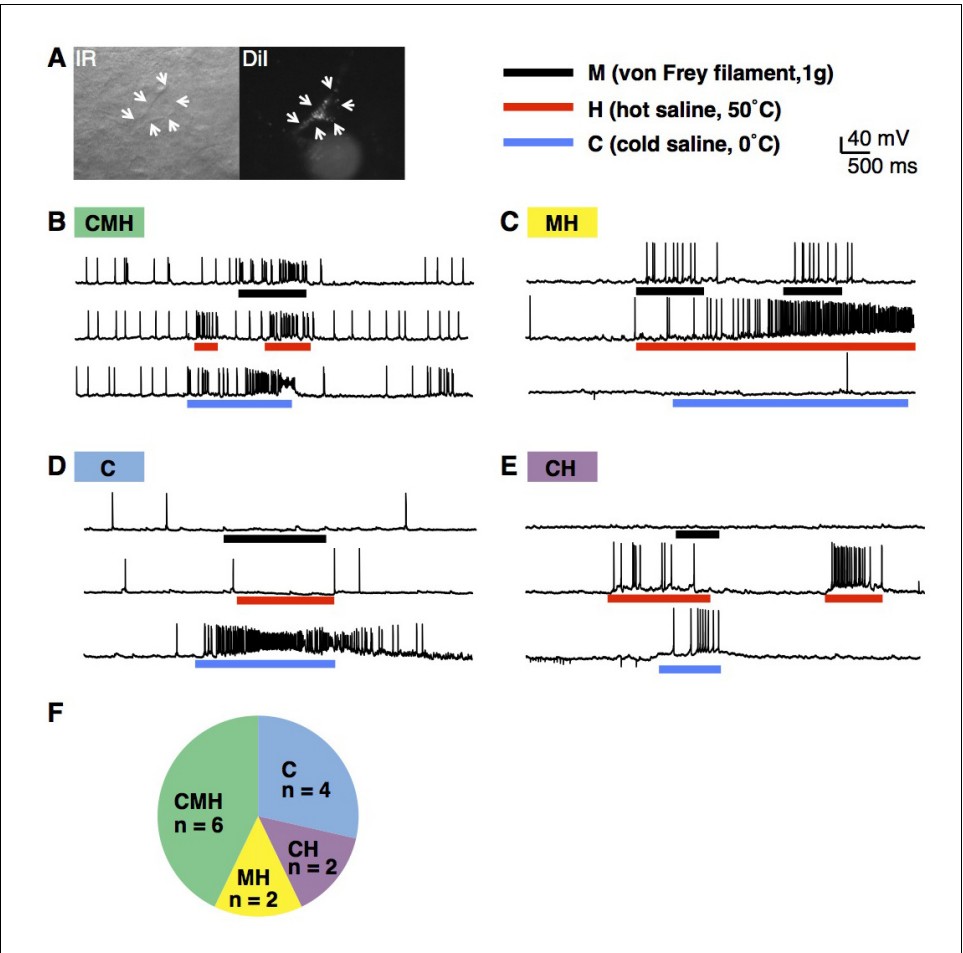

**Figure 3.** Modality tuning of projection neurons. (A) Infrared (IR; left) and epifluorescent image (right) of lamina I projection neuron that was retrogradely labeled by injection of DiI into the lateral parabrachial nucleus. The cell is indicated with arrows. (B–E) Example traces of lamina I projection neurons that respond to mechanical stimulation (M; von Frey filament, 1 g, black bar), heat (H; 50°C saline, red bar) and/or cold (C; 0°C saline, blue bar), (F) Summary of tuning properties of 14 lamina I spinal projection neurons.

throughout the dorsal horn, this also gave us the opportunity to determine the depth from which we could record fluorescent cells in our preparation.

For these experiments, we used the *Bhlhbe22-Cre* allele, which labels neurons that express the transcription factor Bhlhe22 (also known as Bhlhb5) during development. Previous studies from our lab revealed that a subset of Bhlhe22-derived cells in the superficial dorsal horn are inhibitory neurons that function to inhibit itch (*Ross et al., 2010*). We also showed previously, using slice recordings, that *Bhlhe22-Cre* neurons receive direct input from primary afferents that respond to capsaicin, mustard oil, and menthol, raising the possibility that some *Bhlhe22-Cre* neurons might mediate the inhibition of itch by counter-stimuli (*Kardon et al., 2014*). However, whether this primary afferent input was sufficient to cause action potentials in *Bhlhe22-Cre* neurons was unknown.

To address this question, we used mice harboring the *Bhlhe22-Cre* allele together with Ai9, which enables Cre-dependent expression of tdTomato from the *Rosa* locus (here referred to as *Bhlhe22-Cre$^{lsl-tdT}$*). Whole-cell patch-clamp recordings were performed on tdTomato-labeled neurons in the superficial dorsal horn, and the fluorescent dye Alexa 647 was added to the patch pipette to ensure correct targeting (*Figure 4A*). Next, we investigated which types of sensory input (mechanical, cold and/or heat) evoked action potentials in these cells (*Figure 4B*). We found that most *Bhlhe22-Cre* labeled neurons fired in response to at least one stimulus type, and that some responded to all three (*Figure 4C*) as summarized (*Figure 4Ci*; pie chart). Thus, whereas our previous slice recordings had revealed that *Bhlhe22-Cre* cells receive direct input from afferents that respond to counter-stimuli

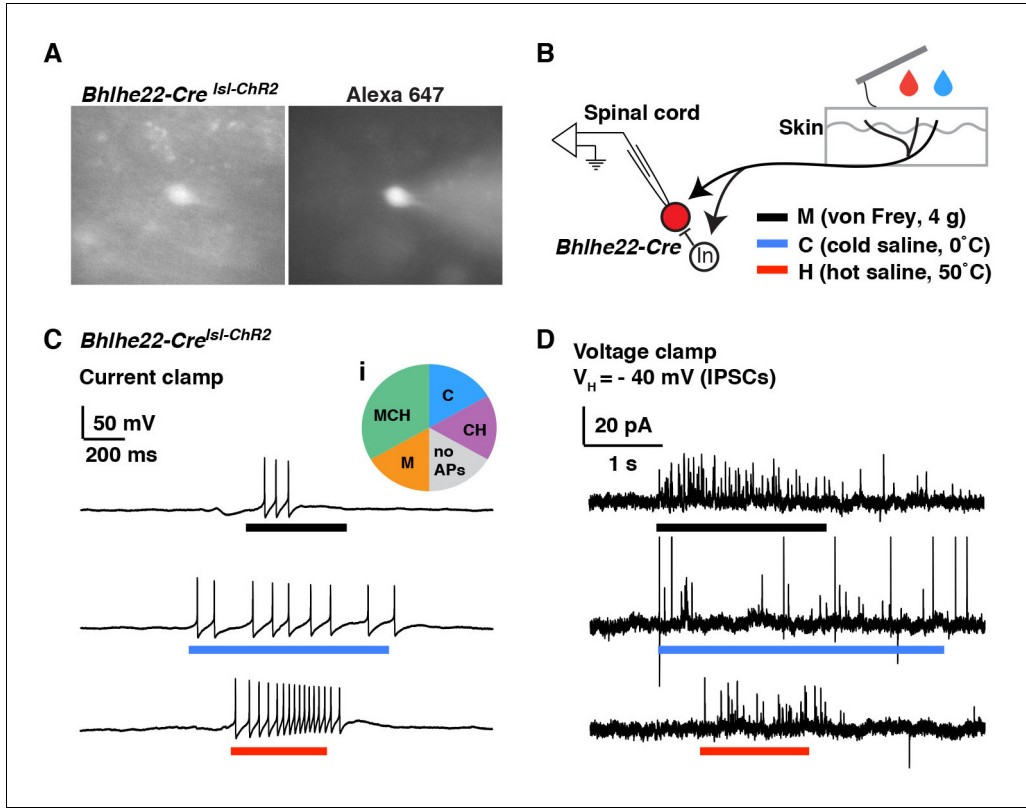

**Figure 4.** Modality tuning (excitatory and inhibitory) of *Bhlhe22-Cre*-labeled spinal interneurons. (**A**) Recording from a lamina II interneuron from a *Bhlhe22-Cre$^{lsl-tdT}$* mouse (left). The cell was filled with alexa 647 to confirm correct targeting (right). (**B**) Diagram illustrating the recording set-up. *Bhlhe22-Cre* cells were characterized based on the types of stimuli that caused action potentials, and the types of stimuli that elicited inhibitory input, as indicated. (**C**) Example traces from a *Bhlhe22-Cre* neuron that showed action potentials in response to mechanical stimulation, cold and heat. Inset (i) shows summary of responses from 6 *Bhlhe22-Cre* neurons. (**D**) Example traces from a *Bhlhe22-Cre* neuron that showed IPSCs in response to mechanical stimulation, cold and heat. C and D are recorded from the same cell.

(*Kardon et al., 2014*), the ex vivo recordings presented here show, for the first time, that *Bhlhe22-Cre* neurons fire action potentials in response to three types of counter-stimuli, heat, cold and mechanical force. Interestingly, we observed that cutaneous stimulation evokes not only EPSCs (data not shown) but also IPSCs in *Bhlhe22-Cre* neurons (*Figure 4D*), suggesting that feed-forward inhibition may gate the responses of *Bhlhe22-Cre* neurons following stimulation of the skin. These data show that it is not only feasible to record from genetically labeled neurons in the semi-intact somatosensory preparation, but that it is possible to characterize their stimulus response properties. Moreover, we could target these brightly fluorescent cells as long as they were within 70 µm from the surface of the spinal cord, suggesting it is possible to study interneurons that reside from lamina I through to the outer part of lamina III.

## Optogenetic modulation of recorded neurons

Another strength of our new semi-intact somatosensory preparation is that it affords the opportunity to map neural circuits using optogenetic approaches. However, to a large extent, these tools have been vetted and optimized in the brain, rather than the spinal cord. There is little precedent for the use of these tools in dorsal horn interneurons (*Cui et al., 2016*; *Foster et al., 2015*), and the degree to which they will work is unclear. We therefore undertook a basic characterization of optogenetic tools in dorsal horn interneurons using several distinct Cre lines and opsins.

For these experiments, we used triple transgenic mice harboring a Cre allele together with two Cre-dependent alleles: Ai32, for expression of ChR2, and Ai9, for expression of tdTomato. The use of tdTomato in these experiments was helpful because, although the ChR2-YFP fusion protein is fluorescent, it remains difficult to resolve which cells express it because this fusion protein is localized to the membrane. In initial experiments, we compared the *Nts-Cre* allele, which is specific to excitatory neurons, and labels approximately half of those in the superficial dorsal horn (*Figure 5—figure supplement 1A*) to the *Bhlhe22-Cre* allele, which mainly labels inhibitory neurons, including those that express Nos1 (also known as nNOS) and/or Galanin (*Chiang et al., 2016*; *Kardon et al., 2014*; *Ross et al., 2010*). Whether recording from cells marked by the *Nts-Cre* allele or by the *Bhlhe22-Cre* allele, we found that blue light induced strong inward currents with amplitudes were not statistically different between populations (*Figure 5A, B and D*). However, it was only in *Nts-Cre* neurons, not *Bhlhe22-Cre* neurons, that blue light resulted in the generation of EPSCs (*Figure 5A*). Furthermore, while the evoked inward current was time-locked to the blue light, the EPSCs in *Nts-Cre* neurons persisted long after the termination of the stimulus. Thus, while the cell autonomous effects of optogenetic activation were similar in *Nts-Cre* and *Bhlhe22-Cre* neurons, the consequences to network activity in these two populations was distinct.

An important factor to consider in these initial optogenetic experiments is that neither the *Nts-Cre* allele nor the *Bhlhe22-Cre* allele is specific to a single neuronal subtype in the dorsal horn. One approach to increase the precision of genetic targeting is through the use of tamoxifen-inducible alleles, which adds specificity through temporal control of recombination. To test the use of optogenetics in this context, we used mice harboring the *Nos1-CreER* allele, which causes recombination selectively in Nos1-expressing neurons (*Figure 5—figure supplement 1B*). For these experiments, mice were treated with tamoxifen at P14 and recordings were performed three weeks later. As before, blue light induced inward current in ChR2-expressing cells (*Figure 5C*); however, the magnitude of the current was significantly lower than that observed using either *Nts-Cre* or *Bhlhe22-Cre* alleles (*Figure 5D*). Given that Nos1 neurons make up a large fraction of those marked by the *Bhlhe22-Cre* allele, it seemed unlikely that this disparity in current was due to a difference in cell type. Rather, we hypothesized that the reduced current in *Nos1-CreER* cells was due to the timing of Cre-mediated recombination, which occurred only upon tamoxifen-treatment, hence allowing less time for ChR2 accumulation. Consistent with this, we found that if we reduced the time post tamoxifen (i.e., one week rather than three), the amplitude of blue light-evoked current was smaller still, and insufficient to trigger an action potential in ChR2-expressing cells (*Figure 5—figure supplement 1C,D and E*). Thus, when using the Ai32 allele, it appears that approximately three weeks' time post-recombination is required to reach functional levels of ChR2 expression in the dorsal horn.

Next, we analyzed effect of stimulation frequency on the responses to optogenetic stimulation. ChR2-expressing *Nts-Cre* neurons were able to follow a low frequency (0.1 Hz; 5 ms), where one (and occasionally two) action potentials were evoked with negligible latency jitter (*Figure 5Ei and F*). In contrast, at higher frequencies (2 and 10 Hz), these neurons could not consistently follow,

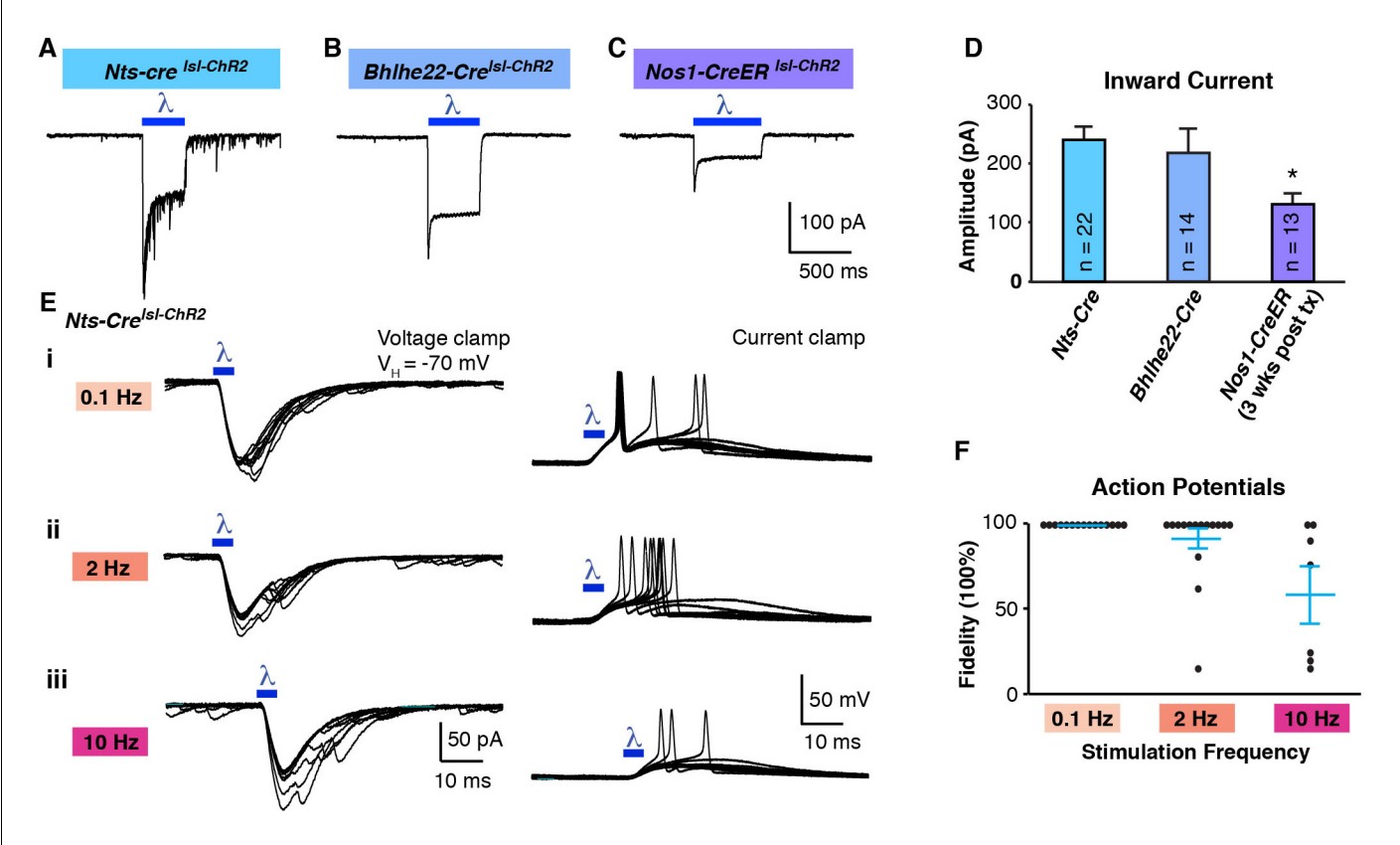

**Figure 5.** Optogenetic activation of spinal interneurons. (A–C) Representative traces of light-induced inward current in *Nts-Cre* (A), *Bhlhe22-Cre* (B) and *Nos1-CreER* (C) neurons expressing ChR2. (D) Quantification of light-evoked inward current. *p<0.05, One-way ANOVA followed by Tukey's test. Data are mean + SEM. Number of recorded cells is indicated. (E) Inward current (left) and action potentials (right) in *Nts-Cre* neurons expressing ChR2 induced by a brief blue light pulse (blue bar; 5 ms) delivered 0.1 Hz (i), 2 Hz (ii) or 10 Hz (iii). Ten traces are superimposed to compare response jitter. (F) Fidelity of optogenetic-induced action potentials in Nts-Cre cells (expressed as percent) varies with frequency of stimulation. Black dots, individual cells; blue lines, mean ± SEM.

The following figure supplement is available for figure 5:

**Figure supplement 1.** Optogenetic activation of spinal interneurons.

demonstrating an increase in jitter and frequent failures (*Figure 5Eii, 5iii and 5F*). Moreover, when similar experiments were performed in ChR2-expressing *Bhlhe22-Cre* neurons, these cells showed even worse fidelity to optogenetic stimulation, and many showed a reluctance to fire even when we increased the stimulus duration (*Figure 5—figure supplement 1F and G*). These data reveal that, while ChR2 can be used in our somatosensory preparation to manipulate the activity of genetically defined cells, the effects of optogenetic stimulation in a given cell type must be determined empirically.

To extend and complement these experiments, we next investigated the degree to which inhibitory opsins could be used to suppress activity in dorsal horn interneurons. For these experiments, we used mice harboring either *Nts-Cre* together with Ai35 (for Cre-dependent Arch) or *Bhlhe22-Cre* together with Ai39 (for Cre-dependent Halo). Expression of either Arch or Halo in dorsal horn neurons enabled light-induced outward current (*Figure 6A, B and C*) and hyperpolarization (*Figure 6D, E and F*) that was not significantly different between *Nts-Cre* and *Bhlhe22-Cre* populations. To determine whether the expression of these opsins was sufficient to block evoked activity, we analyzed the effect of light on activity evoked by electrical stimulation of the dorsal root. In *Nts-Cre* neurons expressing Arch, optogenetic stimulation almost completely blocked root-evoked action potentials (*Figure 6D and G*). In contrast, while the number of root-evoked action potentials was

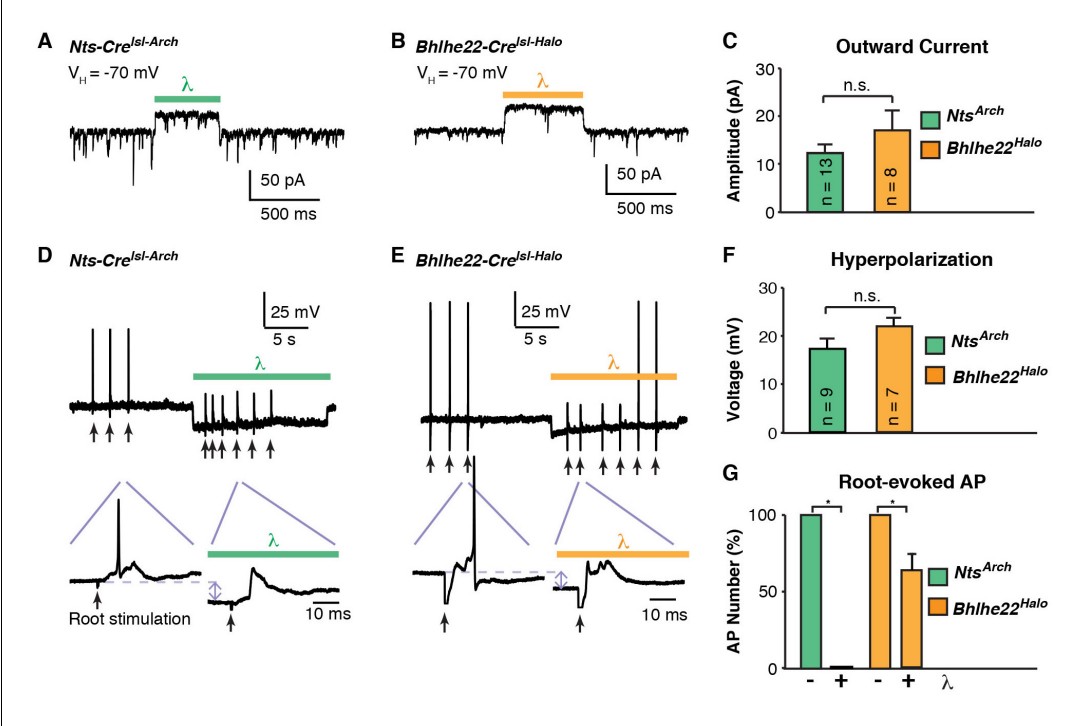

**Figure 6.** Optogenetic inhibition of spinal interneurons. (A–B) Representative traces of light-induced outward current in an *Nts-Cre* interneuron that expresses Arch (A) and a *Bhlhe22-Cre* interneuron that expresses Halo (B). (C) Quantification of optogenetic-induced outward current in *Nts-Cre* neurons expressing Arch (*Nts*^Arch^; green) and *Bhlhe22-Cre* neurons expressing Halo (*Bhlhe22*^Halo^; orange). Data are mean + SEM; n.s., not significant (p>0.05, student's t-test). Number of recorded cells is indicated. (D–E) Representative traces showing the effect of optogenetic stimulation on root-evoked action potentials in *Nts-Cre* neurons expressing Arch (D) and *Bhlhe22-Cre* neurons expressing Halo (E). Root stimulation is indicated with arrows; optogenetic stimulation is marked by bars. Expanded traces are shown below. (F) Quantification of optogenetic-induced hyperpolarization in *Nts-Cre* neurons expressing Arch (*Nts*^Arch^; green) and *Bhlhe22-Cre* neurons expressing Halo (*Bhlhe22*^Halo^; orange). Data are mean + SEM; n.s., not significant (p>0.05, student's t-test). Number of recorded cells is indicated. (G) Quantification of the effect of optogenetic activation on root-evoked action potentials in *Nts-Cre* neurons expressing Arch (*Nts*^Arch^; green) and *Bhlhe22-Cre* neurons expressing Halo (*Bhlhe22*^Halo^; orange). Data are mean + SEM; n = 5 *Nts-Cre* neurons and 3 *Bhlhe22-Cre* neurons; *p<0.05, Student's t-test.

significantly reduced by optogenetic inhibition of *Bhlhe22-Cre* neurons expressing Halo, they were not completely eliminated (*Figure 6E and G*). Given the similarity in outward current and hyperpolarization elicited by the two inhibitory opsins, the disparity in their effectiveness is likely due, at least in part, to differences between the two types of recorded cells, potentially including distinct intrinsic properties and/or differences in root-evoked input. Thus, these findings illustrate that inhibitory opsins can be used in a variety of cell types in the dorsal horn to manipulate cell activity while underscoring the important concept that their effects are context-dependent.

## Circuit mapping using optogenetics

Having determined that we could manipulate neuronal activity in the dorsal horn using optogenetics, we next investigated whether we could utilize these tools to map spinal circuits. In our previous studies, we provided behavioral evidence that a subset of *Bhlhe22-Cre* neurons contribute to the inhibition of itch by counter-stimuli. However, the specific role of *Bhlhe22-Cre* neurons within the context of a spinal microcircuit was unclear. To investigate this question further, we analyzed the effect of optogenetic activation of *Bhlhe22-Cre* neurons on lamina I neurons (*Figure 7A*). Specifically, we searched for lamina I neurons that received direct input from *Bhlhe22-Cre* neurons (*Figure 7B*), which was deemed to be monosynaptic inhibitory input due to the short latency and absence of jitter in the evoked IPSC (data not shown). Next, we analyzed the effect of activating *Bhlhe22-Cre* neurons with optogenetics. In the absence of light, dorsal root stimulation resulted in two phases of EPSCs, likely due to the inputs to this cell from primary afferents with different conduction velocities,

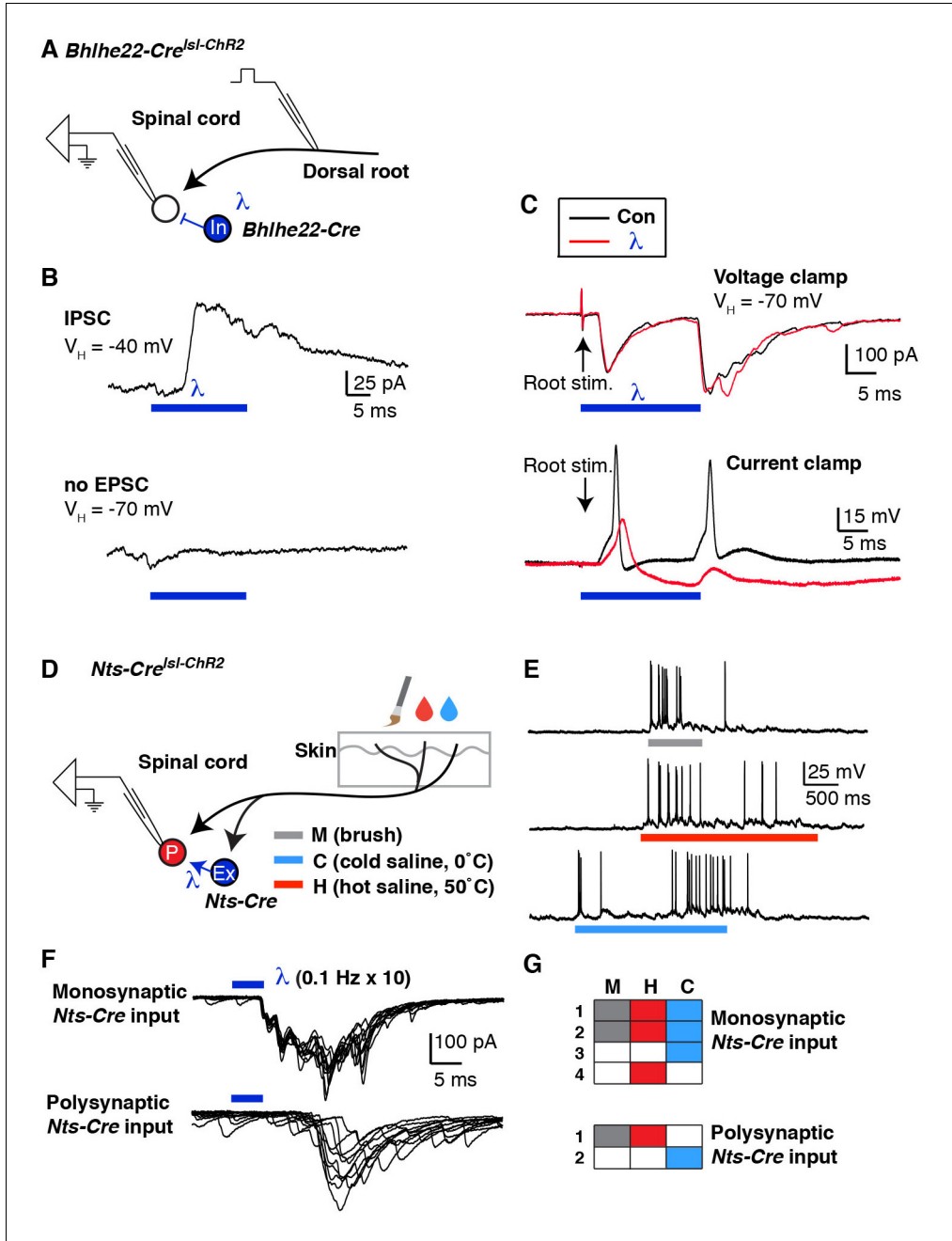

**Figure 7.** Mapping spinal connectivity using optogenetics. (A) Diagram illustrating experimental set-up to test the effect of optogentic activation of *Bhlhe22-Cre* neurons on action potentials mediated by dorsal root stimulation. (B) The recorded cell receives a light-evoked IPSC (top trace) but no light-evoked EPSC (bottom trace). (C) EPSCs (top) and action potentials (bottom) upon electrical stimulation of the dorsal root (arrows) in the absence (black trace) and presence (red trace) of blue light. (D) Diagram illustrating experimental set-up to characterize the modality tuning of spinal projection neurons and characterize whether they receive input from *Nts-Cre* interneurons. (E) Example of a lamina I projection neuron that responds to mechanical stimulation, heat and cold. (F) Representative traces of lamina I spinal projections that receive direct monosynaptic input from *Nts-Cre* cells (top) or only polysynaptic *Nts-Cre* cells (bottom). Ten traces are superimposed to compare response jitter. (G) Modality tuning characteristics of 4 spinal projection neurons that receive monosynaptic *Nts-Cre* input (top) and two spinal projection neurons that receive only polysynaptic *Nts-Cre* input (bottom).

and each of these phases was associated with the generation of an action potential in current clamp (*Figure 7*; black traces). Stimulation of the dorsal root in conjunction with light activation of *Bhlhe22-Cre* neurons resulted in similar EPSCs. However, these EPSCs were no longer associated with the generation of action potentials (*Figure 7C*; red traces). These findings suggest that *Bhlhe22-Cre* neurons can inhibit this recorded cell directly, clamping its activity and preventing it from firing.

Although these experiments demonstrated feasibility, our ability to interpret this experiment was limited by the fact that the identity of the recorded cell was unknown and that the root stimulation was non-physiological. We therefore investigated whether we could map the connectivity of interneurons onto functionally characterized spinal projection neurons. As proof-of-principle, we selected the *Nts-Cre* allele because it is expressed in approximately half of excitatory dorsal horn interneurons. First, DiI-labeled lamina I projection neurons were characterized based on their responses to mechanical, heat and cold stimuli applied to their receptive fields (*Figure 7E*). Then, optogenetic experiments were performed to determine whether the recorded cell received monosynaptic input form *Nts-Cre* neurons (*Figure 7F*). Of six projection neurons that were functionally characterized, only four received monosynaptic input from *Nts-Cre* neurons (*Figure 7G*). However, in this small set of characterized cells we found no evidence that *Nts-Cre* neurons selectively target projection neurons that are tuned to a specific stimulus modality, which may not be surprising given that *Nts-Cre* cells are likely a heterogeneous population. Nevertheless, this experiment illustrates the use of the semi-intact somatosensory preparation together with optogenetic approaches to characterize interneuron input onto functionally characterized spinal projection neurons for the first time. With the use of more selective Cre lines, it seems likely that this approach will help uncover cell type specific connectivity among dorsal horn interneuron subtypes, though whether such connectivity will show modality selectivity remains an open question.

## Discussion

We developed a mouse ex vivo spinal cord preparation for whole-cell patch clamp recording that preserves somatosensory circuits. We revealed that most lamina I neurons have center-surround receptive fields, and that some showed prolonged afterdischarge. In addition, we classified SPB projection neurons based on modality tuning, as well as interneuron connectivity using optogenetics to map spinal circuits. The ability to record from output neurons in the spinal cord while having fine control of input and simultaneously modulating interneuron activity with inhibitory and excitatory opsins should facilitate the functional dissection of spinal microcircuits in the dorsal horn (*Figure 8*).

A key asset of our approach is that it facilitates whole cell patch clamp recordings in the context of a relatively intact circuits, enabling the recording of membrane potentials and synaptic currents, including sub-threshold activity, to characterize both input and output in identified cell types. This

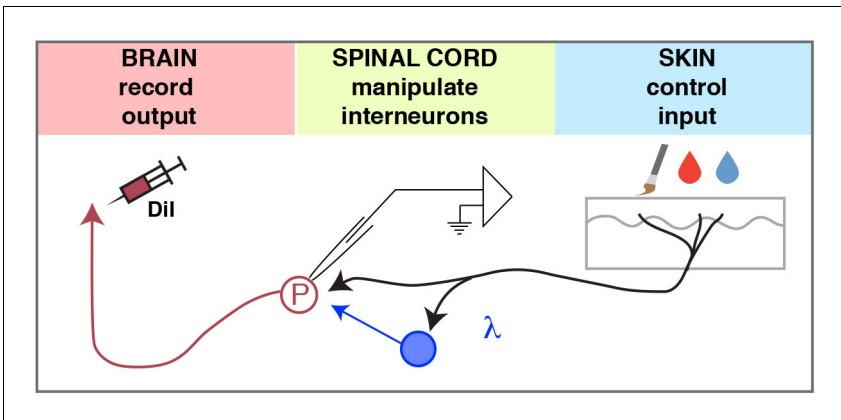

**Figure 8.** Model. The ex-vivo semi-intact somatosensory preparation enables 1) control of the input, 2) manipulation (excitation or inhibition) of spinal interneurons, and 3) quantification of the output to investigate the neural circuits of the dorsal horn.

approach will enable new insight into the circuitry of dorsal horn by assisting in the study of inhibitory mechanisms. Inhibition is thought to be involved in limiting nociception, sharpening spatial acuity, and providing occlusion between somatosensory modalities, all of which become disrupted upon injury (*Lewin et al., 1994*). However, the underlying cellular mechanisms remain largely unknown. Thus, the somatosensory preparation may be a useful tool to study both the neural circuits involved in acute nociception as well as those processes that are disrupted in the context of chronic pain or itch.

A second advantage of this somatosensory preparation over the traditional slice preparation is that it allows the functional characterization of defined cell types in superficial laminae of the dorsal horn, such as spinal projection neurons in lamina I. Understanding the logic of spinal projection neurons that convey noxious information to the brain is a central question in the field that remains unaddressed. In particular, the number of different subtypes of spinal projection neurons is unknown, and how they encode somatosensory information is unclear. For our study, we used a common classification scheme based on three psychophysically distinct modalities, force, heat and cool. A secondary way to classify mechanically sensitive projection neurons is based on the intensity of stimulation to which they respond: nociceptive neurons respond only when mechanical stimulation is noxious, whereas wide dynamic range neurons respond in a graded manner (*Dado et al., 1994*). Most recently, several investigators have begun to subdivide chemically responsive projection neurons based on their responses to chemical algogens and pruritogens (*Andrew and Craig, 2001*; *Carstens, 1997*; *Davidson et al., 2007*; *Jinks and Carstens, 1999*). While these efforts illustrate that different projection neurons are tuned to different types of somatosensory input in complex ways, projection neuron subtypes *per se* have remained generally elusive. In this regard, the semi-intact somatosensory preparation together with development genetic tools that show improved cell-type specificity represent a tool-kit to address this key gap in knowledge.

As with any approach to recording from spinal neurons, ours also has limitations. For example, we can only record from superficial laminae of the spinal cord. In addition, our ex vivo approach involves severing some components, including ascending and descending tracts as well as muscular and visceral inputs. The dissection in itself may cause tissue injury within the vicinity of the receptive fields, potentially altering the properties of the recorded neurons. Could the afterdischarge that we observed in a subset of lamina I neurons occur as a result of tissue damage? While this possibility cannot be excluded at the present time, we think it is unlikely for two reasons. First, afterdischarge is not simply due to injury of primary afferents that could potentially occur while dissecting the skin, since this phenomenon is not observed when we record from DRG neurons in our semi-intact preparation (unpublished observations). Moreover, afterdischarge is also observed in naïve rats upon recording from superficial dorsal horn neurons in vivo, where (just was we report here) 20% of dorsal horn neurons showed prolonged responses to a single stimulus (*Woolf and Fitzgerald, 1983*).

Another potential limitation to the semi-intact somatosensory preparation described here that it enables the recording of inputs from the saphenous nerve (L2 and L3 roots), whereas many behavioral models of pain involve lower spinal segments (L3–L5). There is precedent, however, for the application of test stimuli (e.g., Randall-Selitto apparatus, von Frey filaments) to the dorsal surface of the paw (*Anseloni and Gold, 2008*; *Randall and Selitto, 1957*; *Yilmaz and Gold, 2015*). Furthermore, the spared nerve injury model has been shown to cause mechanical hypersensitivity not only in the sural nerve territory but also in the saphenous nerve territory (*Decosterd and Woolf, 2000*). Therefore, this ex vivo preparation remains compatible with several standard injury models for inflammatory and neuropathic pain.

It is clear that all of our models and circuit diagrams of the dorsal horn are just scratching the surface of the true computations that underlie spinal somatosensory processing. As a first step to understand these computations, it is essential to develop a wiring diagram that clarifies how dorsal horn neurons are connected. Previously, double patch-clamp recordings have been used to investigate synaptic connectivity between interneurons subtypes (*Lu and Perl, 2005*, *2003*). But this approach was limited by the fact that only 10% of the simultaneously recorded pairs of neurons were connected. In contrast, the combination of optogenetics and the ability to record from known cell types in the dorsal horn will allow rapid development of spinal wiring diagrams.

However, our study reinforces the idea that opsin-assisted circuit mapping must be performed with caution because it can be difficult to determine whether synaptic connections are truly monosynaptic owing to the poorer temporal fidelity of optical stimulation. In particular, the typical criteria

used to establish mono-synaptic connectivity using electrical simulation (absence of failure upon stimulation at 20 Hz for Aδ fibers and 2 Hz for C fibers [*Nakatsuka et al., 1999*]) is not applicable to optogenetic stimulation, since light-evoked action potentials have longer latency and show Cre latency jitter than those evoked by electric stimulation (*Figure 5—figure supplement 1F*). For this reason, we and others used a less stringent criterion (e.g., 0.1 Hz optogenetic stimulation) to differentiate between mono- and polysynaptic input in the dorsal horn (*Honsek et al., 2015*; *Wang and Zylka, 2009*; *Cui et al., 2016*). Verification of monosynaptic connections will likely require additional tests involving, for instance, the use of tetrodotoxin together with the potassium channel antagonist 4-aminopyridine (*Nelson et al., 2014*; *Petreanu et al., 2009*).

The idea that neurons in the dorsal horn function as a gate — one that can either block or amplify the nociceptive information that is conveyed to the brain — has been around for decades (*Gebhart, 2004*; *Melzack and Wall, 1965*; *Mendell, 2014*; *Moayedi and Davis, 2013*). To develop novel strategies in the treatment of pain and itch, it is important to understand the neural basis of this gate and the complex computations that are performed within these spinal networks. The somatosensory preparation described here will facilitate these conceptual advances by uncovering spinal somatosensory circuits.

## Materials and methods

### Animals

Four- to seven-week-old mice of both sexes were used in this study. *Bhlhe22-Cre* mice were generated previously (*Ross et al., 2010*); other genetically modified mice were purchased from The Jackson Laboratory, Bar Harbor ME. Two additional Cre alleles were used: *Nts-Cre*, a non-disruptive Cre recombinase knock-in at the endogenous *Neurotensin* locus (stock: 017525); *Nos1-CreER*, a disruptive Cre recombinase knockin to the endogenous neuronal nitric oxide synthase locus (stock: 014541). In addition, we used four Cre-dependent alleles targeted into the *Rosa* locus in which a loxP-flanked STOP cassette prevents CAG promoter-driven expression of reporter and/or effector proteins. These were: Ai9, enabling expression of tdTomato (*Gt(ROSA)26Sor^{tm9(CAG-tdTomato)}*; stock: 007909); Ai32, enabling expression of an enhanced ChR2 fusion protein, ChR2(H134)/EYFP (*Gt(ROSA)26Sor^{tm32(CAG-COP4*H134R/EYFP)}*; stock: 012569); Ai35, enabling expression of an Arch-GFP fusion protein (*Gt(ROSA)26Sor^{tm35.1(CAG-aop3/GFP)}*; stock: 012735); and Ai39, enabling expression of a Halo-GFP fusion protein (*Gt(ROSA)26Sor^{tm39(CAG-hop/EYFP)}*; stock: 014539). Tamoxifen (Sigma, 0.4 mg/kg; IP) was injected into mice harboring the *Nos1-CreER* allele at post-natal day 14 (P14), three weeks prior to electrophysiological experiments. Mice were given free access to food and water and housed under standard laboratory conditions. Wild type mice (C57BL/6) were purchased from Charles River (Horsham, PA). The use of animals was approved by the Institutional Animal Care and Use Committee of the University of Pittsburgh.

### Retrograde labeling of spinal projection neurons

Four- to six-week-old mice were anesthetized with isoflurane and placed in a stereotaxic apparatus. A small hole was made in the skull bone with a dental drill. A glass pipette was used to inject 100 nl of FAST DiI oil (2.5 mg/ml; Invitrogen, Carlsbad, CA) into the left lateral parabrachial area (relative to lambda: anteroposterior −0.5 mm; lateral 1.3 mm; dorsoventral −2.4 mm). The head wound was closed with stitches. After recovery from the anesthesia, the animals fed and drank normally. Mice were used for electrophysiological recordings 4–7 days later.

### Ex vivo somatosensory preparation

Young adult mice (4–7 weeks old) were deeply anesthetized and the hair on the right hindpaw and distal hindlimb was clipped. The animals were perfused transcardially through the left ventricle with ice-cold oxygenated (95% $O_2$ and 5% $CO_2$) sucrose-based artificial cerebrospinal fluid (ACSF) (in mM; 234 sucrose, 2.5 KCl, 0.5 $CaCl_2$, 10 $MgSO_4$, 1.25 $NaH_2PO_4$, 26 $NaHCO_3$, 11 Glucose). Immediately following perfusion, the skin was incised along the dorsal midline and the spinal cord was quickly exposed via dorsal laminectomy. After cord exposure, the right hindlimb and spinal cord (~C2 – S6) were excised, transferred into Sylgard-lined dissection/recording dish, and submerged in the same ice-cold sucrose-based ACSF, which circulated at 50 ml/min to facilitate superfusion of the

cord. Note that this high flow rate did not impede stable recordings because of the chamber volume (20 ml) and architecture (in which the inlet is far from the recording site). Alternatively, a chamber with smaller volume and flow rate requirements can be used for pharmacological studies (*Figure 1—figure supplement 1*). Next, the skin innervated by the saphenous nerve was dissected free of surrounding tissue (modification from *McIlwrath et al., 2007*). L2 and L3 DRG were kept on the spine. Dural and pial membranes were carefully removed and spinal cord was pinned onto the Sylgard block with the right dorsal horn facing upward. Following dissection, the chamber was transferred to the rig. Then the preparation was perfused with normal ACSF solution (in mM; 117 NaCl, 3.6 KCl, 2.5 CaCl$_2$, 1.2 MgCl$_2$, 1.2 NaH$_2$PO$_4$, 25 NaHCO$_3$, 11 glucose) saturated with 95% O$_2$ and 5% CO$_2$ at 32°C. Tissue was rinsed with ACSF for at least 30 min to wash out sucrose. Thereafter, recordings can be performed for up to 6 hr post-dissection.

## Patch clamp recording from dorsal horn neurons

Neurons were visualized using a fixed stage upright microscope (BX51WI Olympus microscope, Tokyo, Japan) equipped with a 40x water immersion objective lens, a CCD camera (ORCA-ER Hamamatsu Photonics, Hamamatsu City, Japan) and monitor screen. A narrow beam infrared LED (L850D-06 Marubeni, Tokyo, Japan, emission peak, 850 nm) was positioned outside the solution meniscus, as previously described (*Safronov et al., 2007*; *Szucs et al., 2009*). Projection neurons in lamina I were identified by DiI fluorescence following injection into the parabrachial nucleus. Whole-cell patch-clamp recordings were made with a pipette constructed from thin-walled single-filamented borosilicate glass using a microelectrode puller (P-97; Sutter Instruments, Novato CA, and PC-10; Narishige International, East Meadow NY). Pipette resistances ranged from six to 12 MΩ. Electrodes were filled with an intracellular solution containing the following (in mM): 135 K-gluconate, 5 KCl, 0.5 CaCl$_2$, 5 EGTA, 5 HEPES, 5 MgATP, pH 7.2. Alexa fluor 647 (Invitrogen; 25 µM) was added to confirm recording from the target cell. Signals were acquired with an amplifier (Axopatch 200B, Molecular Devices, Sunnyvale CA). The data were low-pass filtered at 2 kHz and digitized at 10 kHz with an A/D converter (Digidata 1322A, Molecular Devices) and stored using a data acquisition program (Clampex version 10, Molecular Devices). The liquid junction potential was not corrected.

Root-evoked EPSCs were elicited by stimuli given to the dorsal root (L2 or L3) via a suction electrode. Aδ or C fiber-evoked EPSCs were distinguished on the basis of the conduction velocity of afferent fibers (Aδ > 1.2 m/s; C < 1.2 m/s). Mechanical stimulation was applied using a small paint brush that can activate both nociceptors and non-nociceptors. Thermal stimulation was applied using hot (50°C) and cold (0°C) saline applied gently to the skin using 10 cc syringe and 18 G needle. Von Frey filaments were applied to the skin and held for the duration indicated by the bars.

## Optogenetic activation

Blue light (GFP filter, centered around 485 nm, Lambda DG-4, Sutter instruments) was applied through the objective lens (x40) of the microscope. Light power on the sample was 1.3 mWmm$^{-2}$. The shutter was controlled by Clampex software (Clampex version 10, Molecular Devices). The peak amplitude of inward current induced by blue light stimulation (~500 ms duration) was measured in voltage clamp mode at −70 mV. Connections were considered monosynaptic if the optogenetic stimulus resulted in responses of short (<0.5 ms), invariant latencies following a 5 ms light pulse when stimulated 10 times at 0.1 Hz. To test if activation of *Bhlhe22-Cre* neurons inhibits root-evoked action potentials, root stimulation (0.2 ms) and light pulse stimulation (20 ms) were applied simultaneously. Cy3 cube (centered around 555 nm) was used for activation of Arch and Halo. Peak amplitude of outward current induced by activation of Arch/Halo was measured in voltage clamp mode at −70 mV. To test if activation of Arch/Halo blocks root-evoked action potentials, root stimulation was applied in presence or absence of light stimulation.

## Immunohistochemistry

Young adult mice were deeply anaesthetized and fixed by perfusion with 4% paraformaldehyde. The spinal cord was dissected and post-fixed in 4% paraformaldehyde overnight at 4°C then rinsed thoroughly with 1X phosphate buffer saline (PBS). Transverse sections 60 µm thick were cut with a vibrating microtome from mid-lumbar spinal cord segments (L2 or L3) and processed free-floating for

immunocytochemistry. Sections were washed in 50% ethanol for 30 min, rinsed 4 × 5 min in PBS containing 0.3% Triton X-100 (Sigma, St. Louis MI), and incubated in primary antibodies for three nights at 4°C. These were revealed with species-specific secondary antibodies raised in donkey and conjugated to Alexa fluor 488, Alexa fluor 555, or Alexa fluor 647 (Life Technologies , Carlsbad, CA) diluted 1:500 in and incubated overnight at 4°C. Sections were scanned with a confocal microscope (Nikon AR1) through a 20x or 60x oil-immersion. Scans were analyzed with NeNIS Elements Software (Nikon, Melville NY). Primary antibodies used were chicken anti-green fluorescent protein (Aves Labs, Tigard OR, GFP-1020, 1:2000), rabbit-anti Pax2 (ThermoFisher Scientific, Waltham MA, 71–6000, 1:1000), mouse anti-NeuN (Millipore, Billerica MA, MAB377, 1:1000), and rabbit-anti Nos1 (ThermoFisher Scientific, 61–7000, 1:500).

## Acknowlegdements

We thank Andrew Todd for teaching us how to do the DiI injections into the lateral parabrachial nucleus; MS Gold and NN Urban for critical readings of the manuscript; Research reported in this publication was supported by the National Institute of Arthritis and Musculoskeletal and Skin Diseases of the National Institutes of Health under Award Number R01 AR063772 and R21 AR064445 to SER, as well as the National Institute of Neurological Disorder and Stroke under Award Number NS02372925 to HRK. Part of this work was supported by a grant from the Rita Allen Foundation to SER who is a Rita Allen Foundation Pain Scholar, by NIH grants F31NS092146 and TS NS 73548–3 to LMS and T32 NS073548 to KMB.

## Additional information

### Funding

| Funder | Grant reference number | Author |
| --- | --- | --- |
| National Institutes of Health | AR063772 | Sarah E Ross |
| National Institutes of Health | AR064445 | Sarah E Ross |
| National Institutes of Health | F31NS092146 | Lindsey M Snyder |
| National Institutes of Health | NS735483 | Lindsey M Snyder |
| National Institutes of Health | NS073548 | Kyle M Baumbauer |
| Rita Allen Foundation | | Sarah E Ross |
| National Institutes of Health | NS02372925 | H Richard Koerber |
| National Institutes of Health | NS096705 | H Richard Koerber Sarah E Ross |

The funders had no role in study design, data collection and interpretation, or the decision to submit the work for publication.

### Author contributions

JH, Conception and design; Acquisition of data; Analysis and interpretation of data; Drafting or revising the article; KMB, Acquisition of data; Drafting or revising the article; YO, Conception and design; Acquisition of data; Analysis and interpretation of data ; LMS, Acquisition of data; Analysis and interpretation of data; Drafting or revising the article; HRK, Conception and design; Analysis and interpretation of data; Drafting or revising the article; SER, Conception and design; Analysis and interpretation of data; Drafting or revising the article

### Author ORCIDs

Kyle M Baumbauer, http://orcid.org/0000-0003-0437-9209
Sarah E Ross, http://orcid.org/0000-0003-2593-3133

### Ethics

Animal experimentation: This study was performed in strict accordance with the recommendations in the Guide for the Care and Use of Laboratory Animals of the National Institutes of Health. All of

the animals were handled according to approved institutional animal care and use committee (IACUC) protocol 14043431 of the University of Pittsburgh. All surgery was performed under anesthesia, and every effort was made to minimize suffering.

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
