## [Decision Letter]

Thank you for submitting your article "Semi-intact ex vivo approach to investigate spinal somatosensory circuits" for consideration by *eLife*. Your article has been reviewed by three peer reviewers, and the evaluation has been overseen by Eve Marder as the Senior Editor and Reviewing Editor. The following individuals involved in review of your submission have agreed to reveal their identity: Andrew Todd (Reviewer #1); Carole Torsney (Reviewer #2).

The reviewers have discussed the reviews with one another and the Reviewing Editor has drafted this decision to help you prepare a revised submission.

I am including the full reviews rather than trying to summarize them because they are in essential agreement, and all of them primarily ask for editorial and textual changes. I think it best for you to have the context for the requests for context and more complete descriptions of the limitations of the new method and its potential use. All of the reviewers find your manuscript quite interesting and worthwhile. You will note that you should also be careful to appropriately include citations of relevant previous work, and one of the reviewers is concerned about terminology.

*Reviewer #1:*

The organisation of neuronal circuits that process pain and itch information in the spinal dorsal horn is highly complex. At present, we have very limited information about the responses of different classes of projection neuron or interneuron to natural stimuli, and about the synaptic circuits that these cells take part in. The recent identification of several neurochemically distinct interneuron populations, together with the availability of new mouse lines in which recombinases are expressed in these populations, means that we are now in a far better position to investigate the functional circuitry of pain and itch transmission at the spinal level. However, new techniques will be required for this opportunity to be fully exploited.

The study by Hachisuka et al. makes a major step forwards, by combining 3 different approaches: the skin/nerve/spinal cord preparation developed by Koerber's group, patch-clamp recording in the intact spinal cord with incident LED illumination (developed by Szucs and Safronov), and recombinase-based optogenetic and chemogenetic methods to manipulate activity in genetically-defined neuronal populations. They also show that these techniques can be combined with retrograde labelling to identify projection neurons. Although each of these approaches has been applied to the dorsal horn, the combination is quite novel and provides an extremely powerful way of exploring the functions of different neuronal populations.

My only significant concern is that at various points, the authors refer to the "B5-I neurons", which they have previously defined as inhibitory interneurons in laminae I-II that are dependent on expression of Bhlhb5 for their survival, and correspond to cells that express galanin/dynorphin and/or nNOS. However, in this paper they have a tendency to assume that cells labelled with tdTomato in the *Bhlhb5-Cre;Ai9* cross are the same as the B5-I neurons (e.g. subsection “Recording from genetically labeled neurons”, last paragraph, subsection “Optogenetic modulation of recorded neurons”, second paragraph, and subsection “Circuit mapping using optogenetics”, first paragraph). This is not the case, firstly because some excitatory interneurons will be included, and secondly because there may be inhibitory interneurons that transiently express Bhlhb5, but are not dependent on it for their survival. It would be best to avoid using the term B5-I in this context. In order to restrict the targeted neurons to the B5-I population, they would need some kind of intersectional approach, and that is clearly beyond the scope of this paper.

*Reviewer #2:*

The manuscript by Hachisuka et al. describe a novel ex vivo preparation for the study of spinal somatosensory circuitry in response to natural stimulation of the skin. The successful demonstration of the utility of this approach to study excitatory / inhibitory / subthreshold synaptic input and AP firing in retrograde/genetically defined subpopulations that is evoked by specific sensory modalities is a significant technical achievement and in my opinion would be highly appropriate as a 'tools and resources' article in the *eLife* journal. Furthermore, the authors investigated the use of optogenetic manipulation of particular interneuron subpopulations in this novel preparation as a potentially powerful means of dissecting spinal somatosensory wiring. My main concern with this manuscript is the lack of discussion regarding potential limitations of the preparation and using it in combination with optogenetics to define circuitry. The development of this preparation is commendable and I think that a more open discussion of the potential issues would be a positive asset to such a tools and resources article rather than detract from it.

Key potential issues include the fact that the skin is necessarily cut/incised i.e. there is tissue injury in the vicinity of the saphenous nerve. This is particularly worth mentioning given the demonstration in Figure 2 that a proportion of neurons display 'afterdischarge' – 'afterdischarge' is a feature observed in in vivo recordings from spinal neurons in tissue injury models. Given the paper argues that these approaches will be used to unravel spinal circuitry it has to be acknowledged that, as detailed in the manuscript (Figure 1) this is only achievable within a restricted zone of L2/3 which is accessible for identified patch recording. Furthermore, this has potential implications, if they intend in the future to combine with pain models that classically impact upon L4/5 and affect hindpaw sensitivity.

The most exciting aspect of this study is the potential capacity to combine this preparation with optogenetics to unravel spinal dorsal horn wiring. To define direct/monosynaptic connections high frequency stimulation of inputs is used: monosynaptic/direct connections will be able to reliably follow with little jitter unlike polysynaptic connections which will be variable and or fail. Typically, 20Hz is used in intraspinal studies (with electrical stimulation) to test whether an input is monosynaptic. A prerequisite of this approach is that input must be capable of reliably delivering the repeated stimulus with no jitter. Here the authors demonstrate that with optogenetics this was not straightforward with differing degrees of success between different targeted subpopulations. Their best scenario was 0.1Hz stimulation in the Nts-cre cells (higher frequencies caused significant jitter) – which they then used to define direct connections. This lower stringency could result in false identification of direct monosynaptic connections – this is a caveat that must be discussed if the intention is to use this approach to identify wiring.

*Reviewer #3:*

In this Tools and Resources article Hachisuka et al. combine use of an ex vivo somatosensory system preparation with mouse genetics, retrograde labeling, and optogenetics to clarify the connectivity and response properties of genetically and anatomically defined subsets of neurons in the superficial dorsal horn, including lamina I projection neurons.

Resolving the connectivity and activity patterns of lamina I output neurons is a critically important goal for better understanding how pain / itch information is transmitted to the brain, in health and disease. Developing new methodologies to achieve this goal is crucial to the field.

1) This ex vivo somatosensory system preparation is clearly a very useful approach. The authors convincingly show that this preparation can be combined with mouse genetics, retrograde labelling, and optogenetics, to record from and manipulate the activity of defined populations of dorsal horn neurons. The experiments are generally well executed and controlled, the data support the authors' claims.

2) The ex vivo somatosensory system preparation has been described previously and used extensively by Koerber and colleagues (e.g. Woodbury et al., 2001, doi: 10.1002/cne.1069). While this preparation has been used predominantly to record from DRG neurons, the additions presented here to identify (i.e. mouse genetics, retrograde labeling), and manipulate the activity of dorsal horn neurons with optogenetics, are not particularly innovative but rather commonly used in neuroscience. At the minimum, this should be acknowledged and references to previous studies that have used this preparation, and to optogenetic analysis in the dorsal horn, should be included in the manuscript.

3) A transparent discussion of the limitations of the approach would be a great addition.

The authors describe the preparation as relatively intact, and rightfully so. They indicate its advantages over more severed preparations, such as spinal cord slices. It would be useful to also indicate that important pathways are lesioned in their ex vivo somatosensory system preparation, as these lesions can alter normal/physiological activity and function of the dorsal horn. For example, the axons of lamina I projection neurons the authors record from, and those of all descending control systems that modulate activity of dorsal horn neurons, are cut.

The approach also only permits recording of laminae I-III neurons, and other important output neurons are located in deeper laminae.

The skin is dissected, not intact. Non-neuronal cells in skin are thought to influence activity in somatosensory neurons (e.g. Wilson SR et al., 2013, doi: 10.1016/j.cell.2013.08.057, or this paper from one of the co-authors Baumbauer K et al., doi: 10.7554/*eLife*.09674.): to what extent are these processes impacted by the absence of surrounding tissue? Similarly, the absence of surrounding tissue can alter indentation and transmission of mechanical force.

What advantages does this preparation offer compared with recording of dorsal horn neuron activity in intact animals, for example using calcium imaging (Ran C et al., 2015, doi: 10.1038/nn.4350.), and given the recent development of voltage sensors?

The authors indicate that responses of lamina I neurons are "time locked" to the stimulus duration. This should be rigorously clarified, and it would be useful to know when the sensory terminals are actually depolarized and DRG neurons fire following stimulation, with what delay do lamina I neuron fire, what is the actual force and temperature at the level of the peripheral terminals throughout the stimulation.

---

## [Author Response]

[…]

*Reviewer #1:*

*[…] My only significant concern is that at various points, the authors refer to the "B5-I neurons", which they have previously defined as inhibitory interneurons in laminae I-II that are dependent on expression of Bhlhb5 for their survival, and correspond to cells that express galanin/dynorphin and/or nNOS. However, in this paper they have a tendency to assume that cells labelled with tdTomato in the Bhlhb5-Cre;Ai9 cross are the same as the B5-I neurons (e.g. subsection “Recording from genetically labeled neurons”, last paragraph, subsection “Optogenetic modulation of recorded neurons”, second paragraph, and subsection “Circuit mapping using optogenetics”, first paragraph). This is not the case, firstly because some excitatory interneurons will be included, and secondly because there may be inhibitory interneurons that transiently express Bhlhb5, but are not dependent on it for their survival. It would be best to avoid using the term B5-I in this context. In order to restrict the targeted neurons to the B5-I population, they would need some kind of intersectional approach, and that is clearly beyond the scope of this paper.*

We agree with this comment and now refer to the cells simply as *Bhlhe22-Cre* neurons throughout. (The editors requested that official gene names were used throughout)

*Reviewer #2:*

*[…] My main concern with this manuscript is the lack of discussion regarding potential limitations of the preparation and using it in combination with optogenetics to define circuitry. The development of this preparation is commendable and I think that a more open discussion of the potential issues would be a positive asset to such a tools and resources article rather than detract from it.*

*Key potential issues include the fact that the skin is necessarily cut/incised i.e. there is tissue injury in the vicinity of the saphenous nerve. This is particularly worth mentioning given the demonstration in Figure 2 that a proportion of neurons display 'afterdischarge' – 'afterdischarge' is a feature observed in* in vivo *recordings from spinal neurons in tissue injury models.*

The fact that the tissue is cut is an important point, and we now discuss the possibility that the tissue injury is responsible for afterdischarge as follows:

“As with any approach to recording from spinal neurons, ours also has limitations. […] Moreover, afterdischarge is also observed in naïve rats upon recording from superficial dorsal horn neurons in vivo, where (just was we report here) 20% of dorsal horn neurons showed prolonged responses to a single stimulus (Woolf and Fitzgerald, 1983).”

*Given the paper argues that these approaches will be used to unravel spinal circuitry it has to be acknowledged that, as detailed in the manuscript (Figure 1) this is only achievable within a restricted zone of L2/3 which is accessible for identified patch recording. Furthermore, this has potential implications, if they intend in the future to combine with pain models that classically impact upon L4/5 and affect hindpaw sensitivity.*

The fact that this approach enables recordings from a restricted zone is an important limitation that we now discuss as follows:

“Another potential limitation to the semi-intact somatosensory preparation described here that it enables the recording of inputs from the saphenous nerve (L2 and L3 roots), whereas many behavioral models of pain involve lower spinal segments (L3 – L5). […] Therefore, this ex vivo preparation remains compatible with several standard injury models for inflammatory and neuropathic pain.”

*The most exciting aspect of this study is the potential capacity to combine this preparation with optogenetics to unravel spinal dorsal horn wiring. To define direct/monosynaptic connections high frequency stimulation of inputs is used: monosynaptic/direct connections will be able to reliably follow with little jitter unlike polysynaptic connections which will be variable and or fail. Typically, 20Hz is used in intraspinal studies (with electrical stimulation) to test whether an input is monosynaptic. A prerequisite of this approach is that input must be capable of reliably delivering the repeated stimulus with no jitter. Here the authors demonstrate that with optogenetics this was not straightforward with differing degrees of success between different targeted subpopulations. Their best scenario was 0.1Hz stimulation in the Nts-cre cells (higher frequencies caused significant jitter) – which they then used to define direct connections. This lower stringency could result in false identification of direct monosynaptic connections – this is a caveat that must be discussed if the intention is to use this approach to identify wiring.*

We totally agree that there are caveats to the interpretation the optogenetic experiments. We now discuss this as follows:

“However, our study reinforces the idea that opsin-assisted circuit mapping must be performed with caution because it can be difficult to determine whether synaptic connections are truly monosynaptic owing to the poorer temporal fidelity of optical stimulation. […] Verification of monosynaptic connections will likely require additional tests involving, for instance, the use of tetrodotoxin together with the potassium channel antagonist 4-aminopyridine (Nelson et al., 2014; Petreanu et al., 2009).”

*Reviewer #3:*

*[…] 1) This ex vivo somatosensory system preparation is clearly a very useful approach. The authors convincingly show that this preparation can be combined with mouse genetics, retrograde labelling, and optogenetics, to record from and manipulate the activity of defined populations of dorsal horn neurons. The experiments are generally well executed and controlled, the data support the authors' claims.*

*2) The ex vivo somatosensory system preparation has been described previously and used extensively by Koerber and colleagues (e.g. Woodbury et al., 2001, doi: 10.1002/cne.1069). While this preparation has been used predominantly to record from DRG neurons, the additions presented here to identify (i.e. mouse genetics, retrograde labeling), and manipulate the activity of dorsal horn neurons with optogenetics, are not particularly innovative but rather commonly used in neuroscience. At the minimum, this should be acknowledged and references to previous studies that have used this preparation, and to optogenetic analysis in the dorsal horn, should be included in the manuscript.*

These are valid points. Please note that the preparation described by Woodbury at al. (2001) was one containing dorsolateral trunk skin. The preparation used here, which contains skin of the dorsal hindlimb skin was first described in McIlwrath et al., 2007. We have now clarified this issue as follows:

“Previously, we developed an ex vivo preparation to record from primary afferents while providing natural stimulation to the skin (Baumbauer et al., 2015; Jankowski et al., 2009; Lawson et al., 2008; McIlwrath et al., 2007). Here, we extended the use of this preparation by recording from neurons in the superficial dorsal horn rather than the dorsal root ganglia.”

In addition, we cited two papers (Foster et al., 2015; Cui et al., 2016) in which optogenetics are used to manipulate the activity of interneurons in the dorsal horn.

*3) A transparent discussion of the limitations of the approach would be a great addition.*

*The authors describe the preparation as relatively intact, and rightfully so. They indicate its advantages over more severed preparations, such as spinal cord slices. It would be useful to also indicate that important pathways are lesioned in their* ex vivo *somatosensory system preparation, as these lesions can alter normal/physiological activity and function of the dorsal horn. For example, the axons of lamina I projection neurons the authors record from, and those of all descending control systems that modulate activity of dorsal horn neurons, are cut.*

*The approach also only permits recording of laminae I-III neurons, and other important output neurons are located in deeper laminae.*

These are valid points. We now acknowledge these limitations as follows:

“As with any approach to recording from spinal neurons, ours also has limitations. For example, we can only record from superficial laminae of the spinal cord. In addition, our ex vivo approach involves severing some components, including ascending and descending tracts as well as muscular and visceral input.”

*The skin is dissected, not intact. Non-neuronal cells in skin are thought to influence activity in somatosensory neurons (e.g. Wilson SR et al., 2013, doi: 10.1016/j.cell.2013.08.057, or this paper from one of the co-authors Baumbauer K et al., doi: 10.7554/eLife.09674.): to what extent are these processes impacted by the absence of surrounding tissue?*

Our preparation comprises a large chunk of intact dermis that contains non-neuronal cells of the skin. Indeed, the key role of signaling from keratinocytes to primary afferents in the modulation of afferent threshold was discovered using this somatosensory preparation (Baumbauer et al., 2016). The paper by Wilson et al. (2013) shows that TSLP released from keratinocytes causes itch. The overlying dermal layer (containing keratinocytes and dermal immune cells) is intact in our preparation, thereby enabling, for instance, TLSP released from keratinocytes to activate itch-responsive primary afferents. Thus, we feel that this preparation is a good model with which to study the communication between skin and nerves.

In contrast, the semi-intact somatosensory preparation is not a good model with which to study the possible role of fat, muscle and connective tissue beneath the skin and/or circulating factors on the activity of nerves that innervate the skin. The finding that receptive field properties that we observe using this preparation are similar to those that have been observed in vivo argues that underlying structures do not play a large role, at least not in the context of acute signaling. These ideas could be added to the Discussion, but we have opted not to add them since the Discussion is already very long.

*Similarly, the absence of surrounding tissue can alter indentation and transmission of mechanical force.*

Here we are not sure whether the reviewer in concerned about the absence of surrounding skin, the absence of underlying tissue, or both.

Given the large area of skin surface in our preparation (~7 cm^2^) relative to the small area of a von Frey filament (<0.1 mm^2^), we think it unlikely that the absence of surrounding skin will have a big impact on the transmission of mechanical force.

We agree that the transmission of mechanical force could be affected the rigidity of the material that is beneath the skin. For example, primary afferent responses to a given force may differ depending on whether the skin is atop something rigid (such as bone or a metal platform) vs. something pliant (such as adipose tissue). While worthy of consideration, since the data in our manuscript are qualitative rather than quantitative, we feel that a discussion of these ideas is beyond the scope of this manuscript.

*What advantages does this preparation offer compared with recording of dorsal horn neuron activity in intact animals, for example using calcium imaging (Ran C et al., 2015, doi: 10.1038/nn.4350.), and given the recent development of voltage sensors?*

The reviewer brings up two important considerations. One is the relative advantages/disadvantages of an ex vivo approach compared to an in vivoapproach. A second but distinct consideration is the relative merits of electrophysiological recordings of individual neurons vs. optical monitoring of populations of neurons (note that both of these approaches can be coupled with either in vivoor ex vivo preparations.

Regarding the first consideration (ex vivo vs. in vivo), we discuss the advantages and the limitations of the ex vivo at length in the Discussion.

Regarding the second consideration (electrophysiology vs. optical monitoring), there are clearly pros and cons to each approach. Electrophysiological recordings offer better temporal resolution and the ability to resolve IPSPs, EPSPs and individual action potentials; optical monitoring enables the simultaneous analysis of populations of neurons. Although a combination of these approaches will clearly be important to understand neural circuit function, discussing calcium imaging and the future use of voltage sensors is, in our opinion, outside the scope of the current manuscript.

*The authors indicate that responses of lamina I neurons are "time locked" to the stimulus duration. This should be rigorously clarified, and it would be useful to know when the sensory terminals are actually depolarized and DRG neurons fire following stimulation, with what delay do lamina I neuron fire, what is the actual force and temperature at the level of the peripheral terminals throughout the stimulation.*

We now define what we mean by time-locked in the following manner:

“Most lamina I neurons (48 of 61) showed EPSCs only during the period of mechanical stimulation (time-locked; Figure 2).”

It might be preferable to remove the use of this term altogether. However, we felt it was useful in the figure where we needed a short description that would allow the reader to understand the data in the pie graph.

Since the stimulation was performed manually, we do not have millisecond temporal precision that is afforded by electrically-controlled stimulation. In addition, in these particular experiments we do not know the precise force or temperature at the level of the peripheral terminals. Hence, all of the results described here are qualitative rather than quantitative.